# Study protocol for connective tissue disease-associated interstitial lung disease trial (TEL-CTD-ILD): A randomized controlled trial of a home-based telemonitoring of treatment effects

**Sylwia Małysiak-Szpond**[1], **Maria Mozga**[1], **Ewa Miądlikowska**[1], **Joanna Miłkowska-Dymanowska**[1], **Adam Jerzy Białas**[2], **Wojciech Jerzy Piotrowski**[1]\*

1 Department of Pneumology, I Chair of Internal Medicine, Medical University of Lodz, Lodz, Poland,
2 Department of Pathobiology of Respiratory Diseases, II Chair of Internal Medicine, Medical University of Lodz, Lodz, Poland

\* wojciech.piotrowski@umed.lodz.pl

## Abstract

### Introduction

Interstitial lung disease is one of the most severe pulmonary complications related to connective tissue diseases, resulting in substantial morbidity and mortality. Telepneumology has the potential to improve the long-term management of patients with CTD-ILD. We propose a randomized controlled trial to evaluate the efficacy of home-based telemonitoring of patients with CTD-ILD, in whom treatment was initiated.

### Materials and methods

We will conduct a randomized controlled trial comparing the standard of care with a telemonitoring program. Telemonitoring will start 10 to 14 days before treatment and will be carried out for three months of therapy. After initial training, patients from the intervention group will perform daily spirometry (FVC), transdermal pulse oximetry, pulse and blood pressure measurements, activity measurement (accelerometry), and assessment of the severity of cough and dyspnea. The results will be reported using a telemetric system designed by Mediguard® for this study. The primary outcome measure will be the health-related quality of life change using EQ-5D-5L questionnaire and St. George's Respiratory Questionnaire, as measured at stationary visits in both study groups. Secondary outcomes will include assessment of lung function, costs of health service utilization, satisfaction from being telemonitored, dyspnea by mMRC, fatigue by FAS, patients' adherence to recommended medications using the ASCD, anxiety and depression symptoms as measured by HADS, PHQ-9, and side effects of treatment.

### Discussion

This is the first clinical trial protocol to evaluate home-based telemonitoring to optimize connective tissue disease-associated interstitial lung management. The study aims to provide

**Data Availability Statement:** Deidentified research data will be made publicly available when the study is completed and published.

**Funding:** The study is funded by the Polish Ministry of Science and Higher Education (0047/DW/2018).

**Competing interests:** The authors have declared that no competing interests exist.

data on the impact of telemonitoring on quality of life, evaluation of health status of patients with CTD-ILD using telemonitoring versus standard care. Additionally, we will evaluate the cost-effectiveness of telemonitoring solutions in patients with CTD-ILD.

## Trial registration

ClinicalTrials.gov Identifier: NCT04428957; Registered June 11, 2020; https://clinicaltrials.gov/ct2/show/NCT04428957.

## 1. Introduction

Interstitial lung disease (ILD) is one of the most severe pulmonary complications related to connective tissue diseases (CTDs), resulting in substantial morbidity and mortality. Interstitial lung disease (ILD) and connective tissue disease (CTD) are heterogeneous, complex diseases that require a systematic diagnosis and cross-disciplinary care [1–4]. Approximately 30% of individuals presenting with ILD suffer from CTD. In about 15% of these patients, the development of ILD anticipates the appearance of a complete clinical picture of CTD [5].

Publications show broad ranges in the prevalence of CTD-ILD between countries. Interstitial lung disease is a common manifestation of different connective tissue diseases, such as scleroderma, rheumatoid arthritis (RA), Sjögren's syndrome, systemic lupus, dermatomyositis, and others [6]. ILD is reported in up to 90% of patients with systemic sclerosis, 4–68% of rheumatoid arthritis, 20–85% of mixed connective tissue disease, and 15–70% of the inflammatory myopathies—polymyositis and dermatomyositis [7, 8]. The prevalence of CTD-ILD was approximated as 12.14 per 100.000 [9].

Most often, CTD-ILD patients display radiological and histopathological patterns of non-specific interstitial pneumonia (NSIP), usual interstitial pneumonia (UIP), organizing pneumonia (OP), fibrosing OP, diffuse alveolar damage (DAD), and lymphocytic interstitial pneumonia (LIP).

The current standard of care in progressive non-fibrotic CTD associated ILD is low to medium doses of corticosteroids, frequently combined with immunosuppressive medication, depending on disease severity and local standards. However, based on clinical and radiological features, it is difficult to predict the response to the treatment. The effectiveness of the treatment is assessed by functional tests and chest high resolution computed tomography (HRCT), usually performed after 3 months of therapy [10].

CTD-ILD patients require accurate monitoring of pulmonary complications, symptoms, and pulmonary function trends for optimal long-term management. Home telemonitoring of chronic diseases seems to be a promising patient management approach. According to our knowledge, there are currently no randomized clinical studies published comparing CTD-ILD patients subjected to telemonitoring with a control group consisting of patients under conventional clinical follow-up.

The study protocol objective is to assess the possible benefits of telemonitoring of clinical parameters, symptoms, and quality of life of patients with CTD-ILD in response to treatment. The health-related quality of life is crucial in evaluating the patient-oriented approach to decision-making in the primary outcome measure. We will possibly gain evidence on telemedicine's cost-effectiveness by presenting the results of a cost-utility analysis of a telemonitoring intervention to patients with CTD-ILD compared with the standard of practice.

## 2. Materials and methods

### 2.1. Study principles

The protocol follows the SPIRIT 2013 (Standard Protocol Items: Recommendations for Interventional Trials) and the Template for Interventions Description and Replication (TIDieR) checklist for a description of the interventions [11, 12] (Fig 1). Once completed, the reporting will follow the CONSORT (Consolidated Standards of Reporting Trials) Statement for non-pharmacologic trials [13] (Fig 2). Data collection for the study will be completed by June 2023.

| | Enrolment | Allocation | Post-allocation | |
|---|---|---|---|---|
| **TIMEPOINT** | $-t_1$ | **0** | $t_1$ | $t_2$ |
| **ENROLMENT:** | | | | |
| **Eligibility screen** | X | | | |
| **Informed consent** | X | | | |
| **Allocation** | | X | | |
| **INTERVENTIONS:** | | | | |
| *Telemonitoring program* | | ●————————● | | |
| *Control* | | ●————————● | | |
| **ASSESSMENTS:** | | | | |
| **Primary outcomes:** | | | | |
| Health-related quality of life using EQ-5D-5L questionnaire | | X | X | |
| Health-related quality of life using St. George's Respiratory Questionnaire | | X | X | |
| **Secondary outcomes:** | | | | |
| Costs of health service utilization | | | X | |
| Dyspnea using Modified Medical Research Council (mMRC) | | X | X | |
| Fatigue using Fatigue Assessment Scale (FAS) | | X | X | |
| Patients' adherence using the Adherence Scale in Chronic Diseases (ASCD) | | X | X | |
| Anxiety and depression symptoms as measured by Hospital Anxiety and Depression Scale (HADS) | | X | X | |
| Depression as measured by Patient Health Questionnaire (PHQ-9) | | X | X | |
| **Telemonitoring group** | | | | |
| Lung function | | | | |
| ▪ Oxygen saturation (SpO2) | | ●————————● | | |
| ▪ Heart rate (HR) | | ●————————● | | |
| ▪ Systolic blood pressure (SBP) | | ●————————● | | |
| ▪ Diastolic blood pressure (DBP) | | ●————————● | | |
| ▪ Forced vital capacity (FVC) | | ●————————● | | |
| ▪ Forced expiratory volume in 1st second (FEV1) | | ●————————● | | |
| ▪ Cough severity measured using 5-point Likert scale | | ●————————● | | |
| ▪ Dyspnea severity measured using a 5-point Likert scale | | ●————————● | | |
| ▪ Patient's satisfaction assessed by telemonitoring satisfaction survey | | | X | |

**Fig 1. SPIRIT schedule of enrolment, interventions, and assessments.**

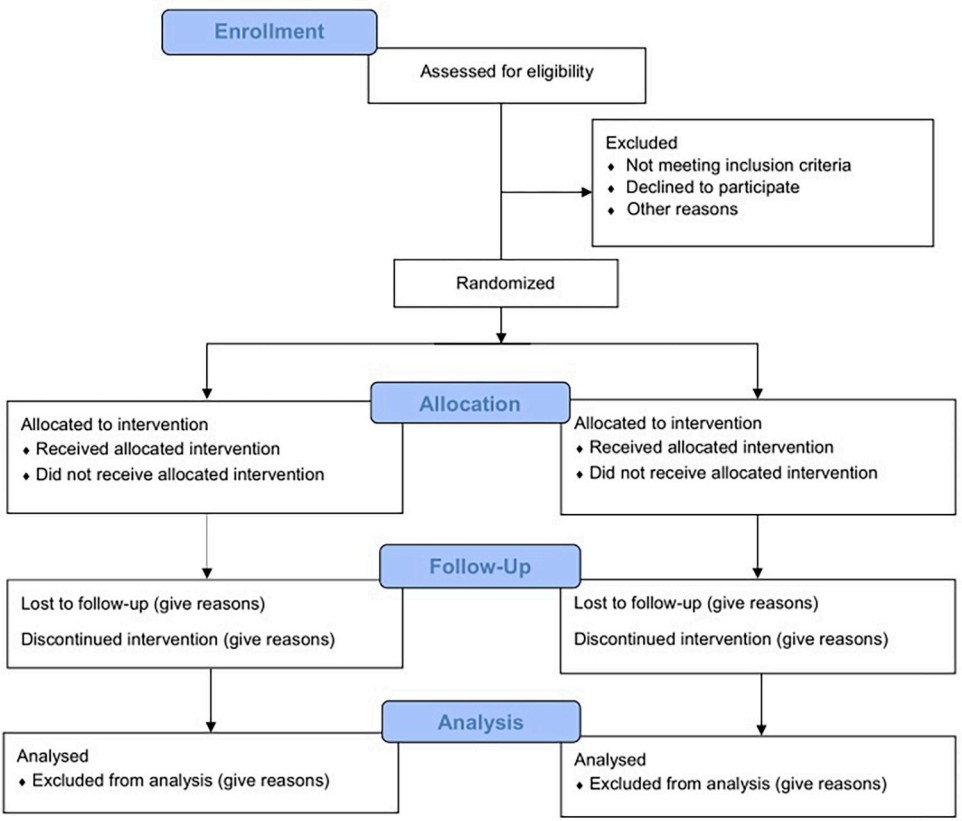

**Fig 2. CONSORT 2010 flow diagram.**

## 2.2. Study design

The study was designed as a single institution randomized controlled trial (Fig 3). The trial investigates the effect of telemonitoring on quality of life. Telemetry will include functional and vital signs, symptoms of patients with CTD-ILD in response to treatment, and potential

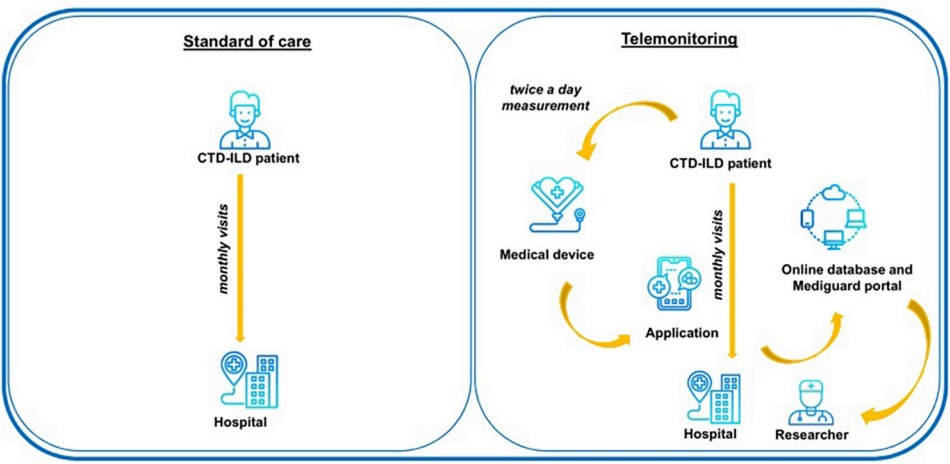

**Fig 3. TEL-CTD-ILD study design.**

side of effects. (ClinicalTrial.gov-identifier: NCT04428957). Active recruitment strategies that engage health care professionals will be applied. Patients of the Department of Pneumology of the Medical University of Lodz or the adjacent outpatient clinic diagnosed with CTD-ILD will be randomized to the intervention group (telemonitoring) or the control group (conventional assessment). The Ethical Committee for Human Studies of the Medical University of Lodz approved the study procedures (No. RNN/88/19/KE). All participants will provide written informed consent at the time of enrolment, and all ethical principles of confidentiality and data protection will be maintained. The protocol of the study was registered in Clinical Trials database at the registration number NCT04428957.

Patients potentially eligible for the study and those who signed the consent will undergo clinical screening: functional tests (spirometry, six-minute walking test, body plethysmography); HRCT examination; bronchofiberoscopy and bronchoalveolar lavage (BAL); transbronchial or transthoracic biopsy if needed; dyspnea and cough assessment at–t1 time point (Fig 1). Primary and secondary outcome measures will be administered at 0 time point (baseline) and t1 time point (12-weeks follow up). The change in health-related quality of life measured by the EQ-5D-5L questionnaire and St. George's Respiratory Questionnaire at three months will be the primary outcome [14–17]. Secondary outcomes are summarized in Fig 1. Patients in the telemonitoring group after initial training will perform daily spirometry (FVC), transdermal pulse oximetry, pulse and blood pressure measurements, activity measurement (accelerometry), and assessment of severity of cough (5-point Lickert scale), dyspnea (mMRC), and fatigue (FAS–fatigue assessment scale) [18–21]. Potential side-effects of treatment will also be assessed. Telemonitoring will start 10 to 14 days before treatment and will be carried out for three months of therapy. All patients will receive treatment following current standards. During the 3-month observation period, visits to the center will take place at monthly intervals. In the case of treatment intolerance or deterioration of monitored parameters, patients will be evaluated at additional time points.

The primary hypothesis is that telemonitoring will affect the quality of life. On the other hand, QoL would measure the treatment effectiveness and be a valuable tool in the healthcare of patients suffering from CTD-ILD. We expect a mean between-group difference of 0.03 score in the EQ-5D-5L test after completing the CTD-ILD telemonitoring programs (primary endpoint at 13-weeks). The minimal clinically significant difference for the EQ-5D-5L is 0.037 ± 0.008 [22]. In the case of the Respiratory Questionnaire of St. George, we assume that the minimum significant difference is 4 units, similar to what was described in COPD [23].

## 2.3. Inclusion and exclusion criteria

Patients eligible for this trial will be newly diagnosed with interstitial lung disease with a less than 10% component of fibrotic changes, will have indications for systemic glucocorticoid therapy +/- immunosuppressant, will be aged 18 years or older. Female patients will be using effective contraception. All patients will need to obtain a positive result in Mini–Mental State Examination ensuring the possibility of efficient operation of monitoring devices and training using telemedicine equipment [24, 25].

Those patients with evidence of irreversible interstitial fibrotic changes in lung HRCT will be excluded. Patients with the pattern of definite or probable UIP in the HRCT examination or contraindications to glucocorticoid and immunosuppressive therapy (azathioprine or mycophenolate mofetil or cyclophosphamide or cyclosporine) will be considered ineligible to participate. Pregnant and breastfeeding women will be excluded from participation in the trial. All study participants will be asked to provide informed written consent to participate in the study.

## 2.4. Randomization

Randomization will be carried out using the envelope method, which will generate the allocation sequence and assign participants to interventions. After baseline assessments, patients will be randomly allocated to the intervention group (telemonitoring) or the control group (standard of care). The randomization will be a 1:1 randomization block from each recruiting.

## 2.5. Sample size

The sample size was determined using data provided by Welling et al [26]. We assume that superiority margin is chosen to be 0.05. Difference in mean SGRQ score is 7.5 and the standard deviation is 15.8. For achieving an 80% power at the 5% level of significance with equal allocation and 10% dropout, the sample size is 26 patients in each group [27]. This calculation was based on taking into consideration quality of life changes as the primary endpoint. To achieve other endpoints higher numbers of examined subjects may be required. Therefore, the first data report will be assumed as preliminary.

## 2.6. Study group

Patients allocated to the control group will receive the usual medical care provided by the health professionals in the Department of Pneumology of the Medical University of Lodz or the adjacent Outpatient Clinic.

   In the home-based telemonitoring group, daily telemonitoring of the following parameters will be obtained: heart rate (HR), blood pressure (BP), pulse oximetry (SpO2), spirometry (FVC), activity (accelerometry), and severity of cough, dyspnea, and fatigue. The results will be reported using telemetric system designed for this study (Mediguard®). In the event of deterioration of monitored parameters below the set thresholds or failure to register them by the patient, team members will be required to contact the patient by phone within the next 24 hours.

   All study patients in the home-based telemonitoring and control group will receive treatment following the current treatment standards. During the 3-month observation period, visits to the center will take place at monthly intervals. In the case of treatment intolerance or deterioration of monitored parameters, patients will be evaluated at additional time points. All patients after the end of the 3-month follow-up will remain under the care of the clinic and will be examined during regular visits every three months until the end of the 12-month follow-up period.

## 2.7. Data analysis

Data analysis will be performed using STATISTICA, StatSoft, Inc. ver. 8.0. statistical package (data analysis software system). In all the calculations, the statistical significance level will be set to $p < 0.05$. Descriptive variables will be presented as means, standard deviation, medians depending on the distribution of the variable. The baseline characteristics of the intervention and control groups will be compared using Student's t-test for normally distributed continuous variables, if the normality can't be assumed Mann-Whitney U test will be used instead. Chi-square test or Fisher's exact test as appropriate for categorical variables comparing groups.

   The primary analysis will be performed for intention to treat (ITT) population. All observed data in primary and secondary outcomes at 3/6 months follow-up will be included in the analysis, with the mixed-effects models. Linear mixed-effects models will be used to handle missing data in the dependent variable using maximum likelihood estimation. Cost-effectiveness analysis will be performed based on cost calculations per quality adjusted life year calculated from

EQ-5D-5 L scores changes over time. Costs and relatives are estimated from national administrative health registries. Costs in telemonitoring and control group related to CTD-ILD treatment and involved in the use of health care services by patients will be estimated based on patients' report questionnaire. Healthcare utilization will be assessed through the number of emergency department, hospital or outpatient clinic visits, medications and adverse events treatments. Resource use categories will be monetarily valued using unit cost and multiplied with the collected amount of resource use.

## 3. Discussion

Implementing appropriate, accessible, affordable, scalable, and sustainable personalized digital health technologies is crucial for WHO Global Strategy on Digital Health 2020–2025 [28]. Digital health can improve health outcomes, efficiency, and cost-effectiveness of health care. The benefit of telemonitoring on detection of pulmonary exacerbations, lung function and quality of life were shown in many studies in chronic lung diseases. Several systematic reviews of telemonitoring in the asthma population concluded that the quality of evidence was inconsistent [29, 30]. Still, the benefits of telemonitoring over usual care in the quality of life were reported. Similarly, based on systematic review data, telemonitoring services in chronic obstructive pulmonary disease created conflicting results due to the high variability of patients monitored, service lines, types of technology, and severity of disease state [31, 32].There are currently no randomized clinical studies published assessing telemonitoring CTD-ILD patients.

The current study will provide a detailed description of the telemonitoring program for patients with CTD-ILD, which aims to provide data on the impact of telemonitoring on quality of life and evaluation of health status of patients with CTD-ILD using telemonitoring versus standard care. The design of the TEL-CTD-ILD trial is based on high-quality criteria of a randomized controlled clinical trial. Mediguard® software used in the trial enables the lung function data to be transmitted automatically via a Bluetooth connection to the researcher's computer. The study team will secure personal data following the highest available standards. The results from this study will most likely provide necessary knowledge regarding treatment response patterns, risk factors for poor response, and evaluation of clinical prognostic factors in patients with CTD-ILD. The study will also contribute to the existing knowledge regarding evaluation of the cost-effectiveness of telemonitoring solutions in patients with CTD-ILD.

Remote techniques of clinical observation are of particular importance in the era of COVID-19 pandemics. In many medical centers, patients encountered difficulties gaining direct contact with their leading physician and other medical staff members in these challenging times. Patients suffering from chronic respiratory diseases are at increased risk of SARS-Cov-2 infection, and the risk increases significantly when immunosuppressive treatment is introduced. In this specific situation, the potential usefulness of telemonitoring application for monitoring treatment effects could reduce the need for on-site visits in a hospital, thus reducing the risk of severe treatment complications.

## Supporting information

**S1 Checklist. SPIRIT 2013 checklist: Recommended items to address in a clinical trial protocol and related documents.**
(DOC)

**S1 File.**
(PDF)

## Acknowledgments

The authors would like to thank the study participants and staff at Department of Pneumology of the Medical University of Lodz for their involvement of this trial.

## Author Contributions

**Conceptualization:** Sylwia Małysiak-Szpond, Maria Mozga, Joanna Miłkowska-Dymanowska, Adam Jerzy Białas.

**Data curation:** Sylwia Małysiak-Szpond, Maria Mozga, Ewa Miądlikowska.

**Funding acquisition:** Sylwia Małysiak-Szpond.

**Investigation:** Sylwia Małysiak-Szpond, Maria Mozga, Ewa Miądlikowska, Wojciech Jerzy Piotrowski.

**Methodology:** Sylwia Małysiak-Szpond, Joanna Miłkowska-Dymanowska, Wojciech Jerzy Piotrowski.

**Supervision:** Adam Jerzy Białas, Wojciech Jerzy Piotrowski.

**Writing – original draft:** Sylwia Małysiak-Szpond.

**Writing – review & editing:** Adam Jerzy Białas, Wojciech Jerzy Piotrowski.

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
