## [Decision Letter · Decision Letter 0]

1 Sep 2022

PONE-D-21-39284Study protocol for Connective Tissue Disease-associated Interstitial Lung Disease Trial (TEL-CTD-ILD): a randomized controlled trial of a home-based telemonitoring of treatment effectsPLOS ONE

Dear Dr. Piotrowski,

Thank you for submitting your manuscript to PLOS ONE. After careful consideration, we feel that it has merit but does not fully meet PLOS ONE’s publication criteria as it currently stands. Therefore, we invite you to submit a revised version of the manuscript that addresses the points raised during the review process.

We look forward to receiving your revised manuscript.

Kind regards,

Minghua Wu, M.D., Ph.D.

Academic Editor

PLOS ONE

Journal Requirements:

3. Thank you for stating the following in the Acknowledgments Section of your manuscript: "The authors would like to thank Polish Ministry of Science and Higher Education for funding support; the study participants and staff at Department of Pneumology of the Medical University of Lodz for their involvement of this trial."

Please remove any funding-related text from the manuscript and let us know how you would like to update your Funding Statement. Currently, your Funding Statement reads as follows: "The study is funded by the Polish Ministry of Science and Higher Education (0047/DW/2018)."

Reviewers' comments:

Reviewer's Responses to Questions

**Comments to the Author**

1. Does the manuscript provide a valid rationale for the proposed study, with clearly identified and justified research questions?

Reviewer #1: Yes

Reviewer #2: Yes

2. Is the protocol technically sound and planned in a manner that will lead to a meaningful outcome and allow testing the stated hypotheses?

Reviewer #1: Yes

Reviewer #2: Yes

3. Is the methodology feasible and described in sufficient detail to allow the work to be replicable?

Reviewer #1: Yes

Reviewer #2: Yes

4. Have the authors described where all data underlying the findings will be made available when the study is complete?

Reviewer #1: Yes

Reviewer #2: Yes

5. Is the manuscript presented in an intelligible fashion and written in standard English?

Reviewer #1: Yes

Reviewer #2: Yes

6. Review Comments to the Author

You may also provide optional suggestions and comments to authors that they might find helpful in planning their study.

Reviewer #1: 1. The sample size calculation does not seem correct. The margin 0.05 is tool small in relative to SD15.8. N 26 should not have enough power.

2. Linear mixed model can only handle missing at random.

Reviewer #2: The authors have created a clear protocol to analyze telemonitoring in patients with CTD-ILD starting treatment compared to standard of care. They have stated clear objectives such as the effect of telemonitoring on quality of life, which in a prior asthma study showed a small improvement, and other punctual secondary objectives which are also important to consider such as cost-effectiveness of the approach. Although not the focus of this study, the results may pave the way for future use of telemonitoring to more closely evaluate treatment response (home vs office based spirometry results and correlation with HRCT chest at follow up).

The protocol is presented clearly, the methods are well described. I suggest, if not already considered, that the treatment each patient receives should be specified and also consider including perceived side effects from treatment as part of patient questionnaires.

7. PLOS authors have the option to publish the peer review history of their article (what does this mean?). If published, this will include your full peer review and any attached files.

Reviewer #1: No

Reviewer #2: No

---

## [Author Response · Author response to Decision Letter 0]

16 Oct 2022

We would like to thank the Reviewers for their input and valuable comments. The answers follow the questions and comments formulated by Editor/Reviewer:

Journal Requirements:

Answer: done according to instruction.

Answer: Grant number and name of funding institution in Financial Disclosure is correct. Potentially confusing information on funding was deleted form the Acknowledgements.

3. Thank you for stating the following in the Acknowledgments Section of your manuscript: "The authors would like to thank Polish Ministry of Science and Higher Education for funding support; the study participants and staff at Department of Pneumology of the Medical University of Lodz for their involvement of this trial."

Please remove any funding-related text from the manuscript and let us know how you would like to update your Funding Statement. Currently, your Funding Statement reads as follows: "The study is funded by the Polish Ministry of Science and Higher Education (0047/DW/2018)."

Answer: We deleted funding information from the Acknowledgments.

Answer: As mentioned in the Material and methods section “Data collection for the study will be completed by June 2023”. The article is a study protocol description; therefore, data are not available at this stage. The answer to the question should be - data availability policy is not applicable to this article.

Answer: the repository information for our data is not yet available. This article does not report data

Answer: not applicable. Reference list is complete and correct.

Reviewers' comments:

Reviewer's Responses to Questions

Comments to the Author

1. Does the manuscript provide a valid rationale for the proposed study, with clearly identified and justified research questions?

Reviewer #1: Yes

Reviewer #2: Yes

Answer: thank you, we appreciate your opinion

2. Is the protocol technically sound and planned in a manner that will lead to a meaningful outcome and allow testing the stated hypotheses?

Reviewer #1: Yes

Reviewer #2: Yes

Answer: thank you, we appreciate your opinion

3. Is the methodology feasible and described in sufficient detail to allow the work to be replicable?

Reviewer #1: Yes

Reviewer #2: Yes

Answer: thank you, we appreciate your opinion

4. Have the authors described where all data underlying the findings will be made available when the study is complete?

Reviewer #1: Yes

Reviewer #2: Yes

5. Is the manuscript presented in an intelligible fashion and written in standard English?

Reviewer #1: Yes

Reviewer #2: Yes

Answer: Thank you, we appreciate your opinion

6. Review Comments to the Author

The Authors would like to thank the Reviewers for thorough assessment and valuable comments. We believe, corrections applied in response to these comments allowed to improve the legibility and quality of our work.

Reviewer #1: 1. The sample size calculation does not seem correct. The margin 0.05 is tool small in relative to SD15.8. N 26 should not have enough power.

Answer: The calculation was done considering quality of life measured by SGRQ, as the primary endpoint of the study. We are aware, that achieving other endpoints may deserve higher numbers of patients. Therefore, first data report will be treated as preliminary, and further studies will be planned, potentially multicenter, to achieve other endpoints.

2. Linear mixed model can only handle missing at random.

Answer: random missing data are expected in this study

Reviewer #2: The authors have created a clear protocol to analyze telemonitoring in patients with CTD-ILD starting treatment compared to standard of care. They have stated clear objectives such as the effect of telemonitoring on quality of life, which in a prior asthma study showed a small improvement, and other punctual secondary objectives which are also important to consider such as cost-effectiveness of the approach. Although not the focus of this study, the results may pave the way for future use of telemonitoring to more closely evaluate treatment response (home vs office based spirometry results and correlation with HRCT chest at follow up).

The protocol is presented clearly, the methods are well described. I suggest, if not already considered, that the treatment each patient receives should be specified and also consider including perceived side effects from treatment as part of patient questionnaires.

Answer: Thank you for your opinion. We plan to describe details of treatment, such as steroid daily dose, type of immunosuppressive drug etc. We also included a questionnaire assessing potential side effects of drugs.

---

## [Editor Report · Decision Letter 1]

21 Nov 2022

Study protocol for Connective Tissue Disease-associated Interstitial Lung Disease Trial (TEL-CTD-ILD): a randomized controlled trial of a home-based telemonitoring of treatment effects

PONE-D-21-39284R1

Dear Dr. Piotrowski,

We’re pleased to inform you that your manuscript has been judged scientifically suitable for publication and will be formally accepted for publication once it meets all outstanding technical requirements.

Kind regards,

Minghua Wu, M.D., Ph.D.

Academic Editor

PLOS ONE
---

## [Editor Report · Acceptance letter]

15 Dec 2022

PONE-D-21-39284R1 

Study protocol for Connective Tissue Disease-associated Interstitial Lung Disease Trial (TEL-CTD-ILD): a randomized controlled trial of a home-based telemonitoring of treatment effects 

Dear Dr. Piotrowski:

I'm pleased to inform you that your manuscript has been deemed suitable for publication in PLOS ONE. Congratulations! Your manuscript is now with our production department. 

Kind regards, 

on behalf of

Dr. Minghua Wu 

Academic Editor

PLOS ONE